# Acute pancreatitis in children – morbidity and outcomes at 1 year

A Bhanot,[1] AA Majbar,[2] Toby Candler  ,[3] LP Hunt,[4] E Cusick,[5] Paul R V Johnson,[6,7] Julian PH Shield  [3,4]

AB and AM are joint first authors.

[1]Faculty of Health Sciences, University of Bristol, Bristol, UK
[2]Department of Paediatrics, Sabratha Teaching Hospital, Sabratha, Libya
[3]Paediatric Diabetes and Endocrinology, University Hospitals Bristol NHS Foundation Trust, Bristol, UK
[4]NIHR Bristol Biomedical Research Centre, Nutrition Theme, University of Bristol, Bristol, UK
[5]Paediatric Surgery, University Hospitals Bristol NHS Foundation Trust, Bristol, UK
[6]Nuffield Department of Surgery, University of Oxford, OXFORD, UK
[7]NIHR Oxford Biomedical Research Centre, University of Oxford, Oxford, UK

**Correspondence to**
Dr Toby Candler; toby.candler@gmail.com

## ABSTRACT

**Objective** To establish short-term and medium-term complications 1-year postdiagnosis, of acute pancreatitis (AP) in children aged 0–14 years.

**Design** One-year follow-up of a prospective monthly surveillance of new cases of AP in children under 15 years through the British Paediatric Surveillance Unit (BPSU) from April 2013 to April 2014.

**Setting** A monthly surveillance of >3700 consultant paediatricians and paediatric surgeons in the UK and Ireland using the BPSU.

**Patients** Children aged 0–14 years with a new diagnosis of AP.

**Main outcome measures** The outcomes following AP, including the incidence of complications and comorbidity at diagnosis and at 1 year.

**Results** Of the 94 new confirmed cases of AP identified in the UK during the study period, 90 cases (96%) were included in the 1-year follow-up. 30 patients (32%) developed further episode(s) of AP. Over one-fifth of patients developed one or more major complication. At initial admission, the most common of these was pancreatic necrosis (n=8, 9%), followed by respiratory failure (n=7, 7%). Reported complications by 1 year were pseudocyst formation (n=9, 10%), diabetes requiring insulin therapy (n=4, 4%) and maldigestion (n=1, 1%). At 1-year postdiagnosis, only 59% of children made a full recovery with no acute or chronic complications or recurrent episodes of AP. Two patients died, indicating a case fatality of ~2.0%.

**Conclusions** AP in childhood is associated with significant short-term and medium-term complications and comorbidities including risk of recurrence in approximately a third of cases.

## INTRODUCTION

Acute pancreatitis (AP) is an inflammatory condition classically presenting with sudden-onset upper abdominal pain.[1] Our previous prospective study found an incidence of AP in UK children under 15 years of 0.78 per 100,000.[2] It is reported that 25% of children develop major complications following an episode of AP.[3 4] The clinical course and complications of AP in children are variable, ranging from mild self-limiting illness to organ failure and death.[5 6]

Pancreatic necrosis, pseudocyst formation and abscesses are some of the complications following AP.[6] Diabetes mellitus (DM) after AP has also been reported in children, with one retrospective study in the USA reporting DM

### WHAT IS ALREADY KNOWN ON THIS TOPIC

⇒ The clinical course and complications of acute pancreatitis (AP) in children are variable, ranging from mild self-limiting illness to organ failure and death.
⇒ Following an episode of AP, children may develop further episodes.
⇒ There has been no prospective study focusing on the complications and morbidities following AP in children and the incidence of these complications.

### WHAT THIS STUDY ADDS

⇒ The most common complications of AP were pancreatic necrosis initially and in the medium term, pseudocyst formation and diabetes requiring insulin.
⇒ Following AP, a significant minority of children will not make a full recovery without residual problems.

### HOW THIS STUDY MIGHT AFFECT RESEARCH, PRACTICE OR POLICY

⇒ Even if classified as mild, childhood AP can lead to significant complications and morbidities and should not be considered a benign condition in childhood.

in 4.5% of cases.[7] Until now in the UK, there has been no prospective study focusing on the complications and comorbidity following AP in children.

Following an episode of AP, children may develop further episodes.[5] Recurrence has been reported in 15%–35% of children with AP.[6] One prospective study in the USA found that 17% of patients developed acute recurrent pancreatitis following an episode of AP.[8] However, the rate of recurrence following AP has yet to be determined in a prospective UK study.

The aim of this study was to conduct a prospective follow-up to our previous national-based analysis of AP in childhood to characterise the outcomes of AP, including determining the prevalence of complications and co-morbidity in the short to medium term.

## METHODS

As previously described, a prospective study was carried out using the British Paediatric Surveillance Unit (BPSU) reporting system

of the Royal College of Paediatrics and Child Health (RCPCH) to identify any new diagnosis of AP in children aged 0–14 years old in the UK and Ireland.[2] From April 2013 to April 2014, AP in children under the age of 15 years was included on the orange card sent monthly to all consultant paediatricians who undertake clinical work in the UK or the Ireland. To identify children directly referred to surgical departments, the British Association of Paediatric Surgeons (BAPS) supported this study and paediatric surgeons received the orange card during the study period.

Diagnosis of AP required at least two of the three following features: (1) Acute onset of upper abdominal pain; (2) Serum amylase and/or lipase raised ≥3 times the upper limit of normal local range and (3) Imaging findings characteristic of AP. These criteria for the diagnosis of AP were adapted according to revised Atlanta criteria by the 'Acute Pancreatitis Classification Working Group' and a consensus of the International Study Group of Paediatric Pancreatitis: In Search for a Cure.[9 10] Clinicians returned the card notifying the BPSU of any AP cases seen or 'nothing to report'. The orange card return compliance was 95.3% during the study.

Each reporting clinician was sent an initial questionnaire to collect additional information on the clinical presentation, diagnosis and management, as well as acute complications and mortality. On the questionnaire's completion, the eligibility of the patient was determined and the physicians' diagnoses were reviewed by the study investigators (AAM, JPHS and EC).[2] If a child had more than one episode of AP during the study period, only the first episode was included as a new case and the information on following episode(s) was used as follow-up data.

A second questionnaire was sent to the reporting clinician 1 year after the initial presentation for diagnostic review, establish management pattern, and identify associated morbidity and outcome. The follow-up questionnaire had four sections. Section A collected information about weight and height. Section B collected data on medium term complications. Section B1 asked about any new surgical interventions, as well as development of pseudocyst, its diagnosis and treatment. Section B2 asked about the development of diabetes, and/or maldigestion and its diagnosis and treatment. Section C detailed further investigations undertaken after initial investigation and Section D asked about recurrence and outcome. The initial questionnaire also collected data on acute complications and mortality.

## Statistics
Data were analysed using the SPSS (V.21.0, IBM), Stata V.12.1 (StataCorp) and Microsoft Excel (2013).

The proportions of patients who were considered to have mild or moderate/severe disease were determined. Factors associated with developing AP were compared between mild and moderate/severe groups using Fisher's exact test.

The percentages of patients developing short-term and medium-term complications up to 1 year postdiagnosis of the first episode of AP were determined. Criteria for considering major complications of the first episode were: (1) all complications required to classify an episode as moderate/severe pancreatitis (organ failure, pancreatic necrosis, and pseudocyst formation), (2) complications requiring treatment at the initial admission (hyperglycaemia) and (3) residual disorders requiring treatment at the end of the follow-up period (diabetes and maldigestion). The major complications were divided into short term (identified early during initial admission) and medium term (identified in patients with prolonged admission or after discharge during follow-up). Minor complications were also reported.

The percentage of patients who developed further episode(s) of AP within 1 year from diagnosis were identified, as well as the average number of recurrent episodes and reported associations.

The prognosis and medium-term outcomes at 1 year of the first episode of AP in children were evaluated by dividing the cohort into five outcome groups: (1) children with full recovery, (2) children with full recovery from the initial episode, but with recurrence of pancreatitis, (3) children with residual problems without recurrence, (4) children with residual problems and recurrence, and (5) children who died while hospitalised.

## RESULTS
From April 2013 to April 2014, there were 94 new confirmed cases of AP identified. At 1-year follow-up, clinical data were available for 90 of these cases (96%), no responses were received for four cases (figure 1).

## Demographics
Female to male ratio was equal with 48/94 (51%) male. The median age at diagnosis was 11.2 years (IQR 7.1–14.4), range 1.3–14.9 years). White children accounted for 57 of 94 cases (61%), compared with 26 (28%) from Asian and f5 (5%) from African ethnic groups. Children of Pakistani origin accounted for the majority (18 of 26; 69%) of the patients of Asian heritage and 19% of the total cohort.[2]

## Outcomes
Sixteen (17%) of the 94 new cases were considered to have moderate/severe disease. There was no association found between aetiology of AP and disease severity (p=0.22, table 1). Two children (2.1%) died in hospital during their first admission: both had underlying chronic conditions (propionic acidaemia and cerebral palsy, global developmental delay, epilepsy and microcephaly). All children required hospital management; eight requiring intensive care. Eighty-six patients (91%) were treated conservatively during their initial hospitalisation. Twenty (21%) required one or more related surgical intervention during the initial admission and/or

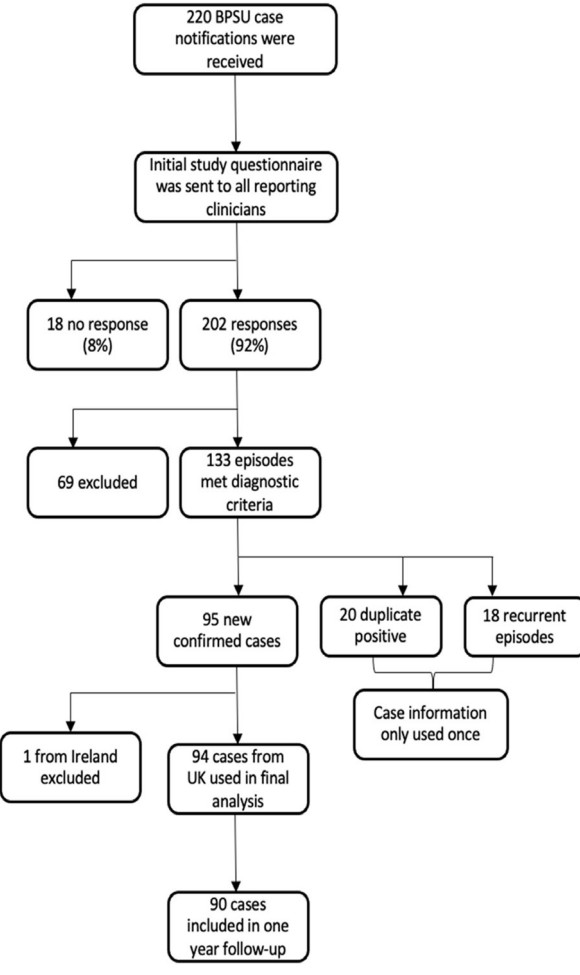

**Figure 1** Recruitment and inclusion process for new acute pancreatitis cases. BPSU, British Paediatric Surveillance Unit.

the follow-up period. The most common indications for surgical intervention were: diagnostic laparoscopies in seemingly idiopathic cases, elimination of the underlying

**Table 1** Main associations of acute pancreatitis in severe and mild cases

| | Severity | |
| | Moderate/severe (n=16) | Mild (n=78) |
| Main association | No of cases (%) | No of cases (%) |
|---|---|---|
| Idiopathic | 4 (25) | 31 (40) |
| Drugs | 6 (38) | 12 (15) |
| Gallstones | 1 (6) | 11 (14) |
| Hereditary | 1 (6) | 6 (8) |
| Organic acidaemia | 2 (13) | 5 (6) |
| Anatomical anomalies | . | 5 (6) |
| Viral infections | 1 (6) | 2 (3) |
| Systemic diseases | . | 2 (3) |
| Trauma | 1 (6) | . |
| Others | . | 4 (5) |

**Table 2** Surgical and interventional procedures performed in 20 patients

| No | Gender, age (years) | Association | Intervention(s) |
|---|---|---|---|
| 1 | Male, 2.3 | Methylmalonic acidaemia | 1. ERCP sphincterotomy* |
| 2 | Female, 12.6 | Gallstones | 1. ERCP sphincterotomy and stones removal*<br>2. Elective cholecystectomy* |
| 3 | Male, 13 | Gallstones | 1. Elective cholecystectomy* |
| 4 | Female, 13.7 | Pancreas divisum | 1. ERCP, pancreatic duct drainage, sphincterotomy and stent insertion* |
| 5 | Male, 14.1 | Sodium valproate | 1. Percutaneous drainage of Left sided abdominal collection* |
| 6 | Male, 14.8 | Gallstones | 1. Laparoscopic cholecystectomy† |
| 7 | Female, 8 | Choledochal cyst | 1. Excision of type I fusiform choledochal malformation* |
| 8 | Male, 4.9 | Idiopathic | 1. ERCP, cyst-duodenostomy stent (Stent removed)* |
| 9 | Male, 10.3 | Trauma | 1. Endoscopic stent for pseudocyst† (unsuccessful)<br>2. Laparoscopic cystgastrotomy for pseudocyst† |
| 10 | Female, 13.5 | Gallstones | 1. Elective laparoscopic cholecystectomy* |
| 11 | Male, 2 | Asparaginase | 1. Diagnostic laparoscopy† |
| 12 | Female, 14.8 | Gallstones | 1. Elective cholecystectomy* |
| 13 | Female, 13.5 | Gallstones | 1. Cholecystectomy and Splenectomy* |
| 14 | Male, 14.6 | Gallstones | 1. ERCP sphincterotomy and stones removal†<br>2. Laparoscopic cholecystectomy* |
| 15 | Male, 5.5 | Choledochal cyst | 1. Excision of choledochal malformation with hepaticojejunostomy* |
| 16 | Female, 12 | Viral infection | 1. Laparotomy and drainage of large peripancreatic collection in lesser sac† |
| 17 | Male, 5.4 | Hereditary | 1. ERCP and stent placement†<br>2. ERCP and stent removal* |
| 18 | Female, 12 | Idiopathic | 1. Diagnostic laparoscopy for acute abdomen† |
| 19 | Male, 14.7 | Gallstones | 1. Elective laparoscopic cholecystectomy* |
| 20 | Male, 14.6 | Gallstones | 1. ERCP and removal of 6 CBD stones† |

*Intervention during the follow-up period.
†Intervention at initial admission.
CBD, common bile duct; ERCP, endoscopic retrograde cholangiopancreatography.

predisposing factors such as gallstones and choledochal cysts, and treatment of complications (table 2).

The outcomes of the 90 cases included at 1 year are shown in table 3. Residual comorbidities included

**Table 3** Outcomes of 90 cases reported at 1 year following acute pancreatitis

| Outcome | No of cases (%) |
|---|---|
| Full recovery | 53 (59) |
| Full recovery from the initial episode but with recurrence of pancreatitis | 26 (29) |
| Residual problems without recurrence | 5 (6) |
| Residual problems with recurrence | 4 (4) |
| Died in hospital | 2 (2) |

insulin-dependent diabetes (n=4), clinical and imaging features of chronic pancreatitis (n=3), maldigestion with failure to thrive (n=1) and residual radiological changes in the pancreas without significant clinical findings (n=1).

### Recurrent disease
Thirty patients (33%) developed further episode(s) of AP within 1 year. These 30 had 88 episodes during the study period with an average of 2.9 episodes/patient (table 4). Twenty-five of these (83%) had mild initial episodes.

The reported aetiology of AP in the recurrent group were idiopathic (n=13, 43%), genetic (n=5, 17%), drugs (n=5, 17%), organic acidaemias (n=4, 13%), pancreas divisum (n=1, 3%), gallstones (n=1, 3%) and α1-antitrypsin deficiency (n=1, 3%). Each of the five drug-associated cases (two asparaginase, one mercaptopurine, one carbamazepine and one opiates) developed one further episode. The recurrent episode in the child whose first episode was due to opiates was again attributed to opiates. The new episode in the child whose first episode was mercaptopurine-associated was ascribed to prednisolone treatment for acute lymphoblastic leukaemia. For the other three cases, available data did not confirm or discount whether the new episodes developed by reintroducing the same medication.

At least one imaging examination was performed during the first episode in the majority of this recurrent group (28 of 30; 93%). Further investigations, including imaging examinations and/or genetic testing, were performed for 22 of the 30 recurrent cases (73%). The genetic and α1-antitrypsin deficiency cases were identified by additional investigations after developing further episode(s). Apart from organic acidaemia cases, all cases with ≥2 recurrent episodes underwent one or more further investigation to exclude possible underlying aetiologies.

**Table 4** Number of episodes of acute pancreatitis in the 30 patients who had a recurrence

| No of episodes | No of children (%) |
|---|---|
| 2 | 17 (57) |
| 3 | 4 (13) |
| 4 | 4 (13) |
| 5 | 4 (13) |
| 6 | 1 (3) |

**Table 5** Major reported complications of acute pancreatitis

| Complication | No of cases* (%) |
|---|---|
| Short term (at initial admission, n=94) | |
| Pancreatic necrosis | 8 (9) |
| Respiratory failure | 7 (7) |
| Hyperglycaemia required insulin therapy | 6 (6) |
| Circulatory collapse | 4 (4) |
| Renal failure | 2 (2) |
| Gastrointestinal bleeding | 2 (2) |
| Medium term (n=90) | |
| Pseudocyst formation | 9 (10) |
| Diabetes | 4 (4) |
| Maldigestion | 1 (1) |

*Some patients had more than one complication.

Genetic screening for SPINK1 (serine peptidase inhibitor, Kazal type 1), PRSS1 (cationic trypsinogen gene) and CFTR (cystic fibrosis transmembrane conductance regulator) gene mutations was performed for 12 of 30 cases with recurrent AP (40%). Of those who had genetic testing, five tests were positive (42%): three SPINK1 and two PRSS1 mutations. Additional imaging identified one case of pancreas divisum. The gallstone-associated case was diagnosed at the first episode, the patient developing a further episode while awaiting surgery. Three children of the 30 with recurrence (10%) showed imaging and clinical signs of chronic pancreatitis.

### Major complications
Acutely or within 1 year, 21/90 children (23%) developed one or more major complication (table 5). Some children developed more than one: one with five complications, four having four, one had three and five had two. Ten children developed one major complication. If recurrence were added as a significant AP complication, the number developing complications would increase to 45 of 90 children (50%).

The most common significant complication reported at initial admission was pancreatic necrosis in eight (9%). The aetiological associations of AP in pancreatic necrosis cases were idiopathic (n=3), medication (two asparaginase, one sodium valproate-associated), gallstones (n=1) and viral (n=1). Three of these eight cases required intensive care therapy. All cases, except one, were treated conservatively. One child died within 10 hours of admission to paediatric intensive care unit without surgical intervention.

Other major complications included respiratory failure in seven (7%), hyperglycaemia which required insulin therapy at some stage in six (6%), circulatory collapse in four (4%), renal failure in two (2%), and gastrointestinal bleeding in two (2%).

The main reported medium-term complications were pseudocyst formation (n=9, 10%), diabetes requiring insulin therapy at 1 year (n=4, 4%) and maldigestion

(n=1, 1%). The aetiological associations of AP in the nine pseudocyst cases were drug therapy (n=4) and one each for trauma, gallstones, propionic acidaemia, viral infection and idiopathic. Six of nine pseudocysts spontaneously resolved during follow-up; two had incomplete follow-up data and one required surgical intervention. One of the diabetes cases had no reported hyperglycaemia initially. The case of maldigestion was diagnosed by both abnormal stool elastase and failure to thrive.

Additional reported acute complications were hypocalcaemia (n=23, 24%), hypoalbuminaemia (n=49, 52%), moderate amounts of free fluid collection in the abdomen (n=28, 30%), pleural effusion (n=13, 14%), lung consolidation (n=3, 3%), lung collapse (one case), deterioration of pre-existing renal dysfunction (n=2, 2%), superior mesenteric vein thrombosis (n=1, 1%) and decompensation of propionic acidaemia (n=1, 1%). Increased seizure activity developed in a child taken off sodium valproate. An episode of AP altered future planned chemotherapy treatment for all children with leukaemia.

## DISCUSSION

This study demonstrates that AP associates with significant morbidity in the short and medium term carrying a small but significant mortality risk. Other than recurrence, over one-fifth of patients develop one or more significant complications. This recapitulates the 20%–25% reported in other previous paediatric studies.[3 4 11] As in several previous studies, pseudocyst formation was the most common major complication, identified in 10% in this current study, the majority of which resolved spontaneously within 1 year.[3 12 13]

Pancreatic necrosis, organ failure and hyperglycaemia requiring insulin treatment are the most frequent significant complications at first admission. In this study, pancreatic necrosis was reported in 9% of cases which was more than the 4.6% reported by Fonseca Sepúlveda and Guerrero-Lozano.[14] This study also shows that pancreatic necrosis may occur with variable aetiological associations in agreement with Raizner et al.[15] Transient hyperglycaemia is quite frequent in children with AP, with six requiring insulin therapy in hospital. Four children (4%) developed diabetes requiring regular insulin treatment at 1 year. This figure is similar to the 4.5% reported by a retrospective study from the United States but less than the 8% reported in a previous UK study.[7 16]

Within the follow-up period, one child with methylmalonic acidaemia developed maldigestion and failure to thrive following a first mild episode of AP. However, reporting of only one patient with pancreatic exocrine insufficiency (PEI) is likely to be a significant underestimate, as several patients had endocrine insufficiency at diagnosis and 1 year later. In adults, it has been found that PEI occurs frequently in individuals recovering from severe AP.[17 18] Studies suggest an absence of clinical symptoms does not allow the prediction of a normal nutritional status or sufficient treatment in chronic pancreatitis patients.[19 20] Therefore, routine evaluation of exocrine pancreatic function using faecal elastase-1

is recommended after all cases of AP to detect any PEI and provide guidance in decisions on enzyme replacement therapy.[21–24]

This study also shows that hypocalcaemia is common acutely, particularly in severe cases, consistent with other reports.[11] No cases required treatment. Other common abnormalities not requiring specific interventions were hypoalbuminaemia (52%), abdominal free fluid collections (30%) and pleural effusions (14%). Future research may benefit from focusing on these common abnormalities and particularly their roles in the prediction of AP severity.

Recurrent pancreatitis is a problem following the first diagnosis of AP. One third of children developed recurrent episode(s) within 1 year, which is comparable to the 28% reported in Italy, 33% in Canada and 36% in Mexico.[25–27] However, it is higher than the 10% previously reported in Scotland, likely due to the significant change in the aetiological associations of AP in childhood, and also higher than the 17% reported from the USA.[2 8 28]

Idiopathic (43%), hereditary factors (17%), medication (17%) and organic acidaemias (13%) are the the most common associations of acute recurrent pancreatitis, similar to findings in other studies.[26 27 29–31] Some of the idiopathic cases may represent under-investigation for genetic or autoimmune causation. This study found relapse may follow both mild and moderate/severe initial AP episodes. However, the majority (83%) of patients who developed recurrence had mild first episodes. When combining both complications and recurrence, 50% (45/90) of the patients in this study developed either major complication(s) from the first episode and/or recurrent pancreatitis within a year from diagnosis. The high recurrence rate with AP observed in this study and several previous paediatric series suggests AP is not a benign disease, even if classified initially as mild.[8 26 27 32]

Two patients died which indicates that AP has a case fatality of ~2.0% in the UK. This is less than the 5.3% and 6.3% reported in two earlier UK series.[16 33] Case fatality figures correspond to the 2% reported by a Scottish report being in line with a review suggesting mortality is less than 5% in most cohorts.[6 28] It is noticeably lower than the 10% reported in adults.[34]

A study limitation was the possibility that diabetes in some cases was the result of a contemporaneous onset of autoimmune type 1 diabetes but data on antibody status was not collected in the survey. Reporting of this follow-up paper was delayed due to the lead researcher being caught up in civil conflict, but we believe the data interpretation is still robust. Patients were reported by paediatricians or paediatric surgeons who are members of BAPS. We believe that the vast majority of cases in those <15 years will have been managed by these professionals. However, there may be a small proportion of patients who are managed solely by adult surgeons (or paediatric surgeons who are not members of BAPS) and would therefore be missed from reporting. Furthermore, a degree of under-reporting is likely with any voluntary surveillance system.

## Open access

In summary, this study suggests that AP is associated with significant medium-term morbidity. One year after diagnosis, only 59% of children made a full recovery with no acute or chronic complications or recurrent episodes of AP. Recurrence and/or other complications were experienced by 41% of patients. Residual comorbidities were identified in about 1/10th of children. Even if classified initially as a mild episode, the prognosis for AP should be guarded.

**Acknowledgements** We need to acknowledge the significant contribution of Mr Richard Lynn, late of the RCPCH BPSU to the successful running of this study. This is an independent opinion from Bristol Biomedical Research Centre in the National Institute for Health Research Biomedical Research Centre and Unit Funding Scheme.

**Contributors** AB wrote the preliminary draft paper under supervision from TC, AAM conducted the BPSU study and performed the data analyses, JPHS developed the mechanics of the study and is the study guarantor, LPH supported and contributed to the statistical analysis, EC and PRVJ edited the questionnaires and took part in diagnostic meeting sessions. All authors contributed to the final editing.

**Funding** This study was funded by the Libyan Ministry of Higher Education and Scientific Research represented by the Libyan Embassy in London as an MD scholarship for Dr A. Majbar at the University of Bristol. Additional funding came from the National Institute for Health Research Biomedical Research Centre Funding Scheme.

**Disclaimer** The views expressed in this publication are those of the authors and not necessarily those of the NHS, the National Institute for Health Research or the Department of Health and Social Care.

**Competing interests** No, there are no competing interests.

**Patient and public involvement** Patients and/or the public were not involved in the design, or conduct, or reporting, or dissemination plans of this research.

**Patient consent for publication** Not applicable.

**Ethics approval** The study was approved by the National Research Ethics Service Committee South West, Central Bristol (REC reference 11/SW/0132) and was granted Section 251 of the National Information Governance Board for Health and Social Care permission by its Ethics andConfidentiality Committee under reference ECC 6-02(FT12)/2012.

**Provenance and peer review** Not commissioned; externally peer reviewed.

**Data availability statement** Data may be obtained from a third party and are not publicly available.

**ORCID iDs**
Toby Candler http://orcid.org/0000-0002-4587-8744
Julian PH Shield http://orcid.org/0000-0003-2601-7575

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
