## [Reviewer comments · BMJ Paediatrics Open]

ARTICLE DETAILS

TITLE (PROVISIONAL)	Long term morbidity and outcomes of acute pancreatitis in children
AUTHORS	Bhanot, A Majbar, AA Candler, Toby Hunt, LP Cusick, E Johnson, Paul R V Shield, Julian PH

VERSION 1 – REVIEW

REVIEWER	Reviewer name: Dr. Emre Basatemur Institution and Country: Barts Health NHS Trust, United Kingdom of Great Britain and Northern Ireland Competing interests: None
REVIEW RETURNED	24-May-2022

GENERAL COMMENTS	This paper describes the clinical outcomes (initial and medium term to 1 year) from a national prospective surveillance study of acute pancreatitis in UK children conducted through the BPSU surveillance system (with additional reporting by paediatric surgeons affiliated with BAPS). The manuscript supplements previously published data from this study, which reported incidence rates, demographic factors, and reported aetiological associations. The paper reports original results, which are of clinical relevance and interest, and merits publication. The authors should be commended on achieving an impressive response rate for 1 year follow-up data (96%). I have a few minor grammatical comments only: 1) There are a few references in the manuscript to 'long term' outcomes. One year follow-up would more appropriately be described as 'medium term' rather than long term. References to 'long term' should be changed to 'medium term'. E.g. in the short title (p1), 2nd line of the discussion (p12)2) The text referring to the results in table 1 (page 8, outcomes subsection, 2nd sentence) could be written more clearly. I suggest something along the lines of: 'There was no association found between aetiology of AP and disease severity (p=0.22, Table 1).' Also, on p9 line 28, I suggest changing the text from 'The reported associations of AP...' to 'The reported aetiology of AP....'3) The authors report that 21% of cases required a surgical intervention. Do the authors have further information available regarding the types of surgical procedures performed?4) In the 1st paragraph of the discussion, the last sentence concerns pseudocyst formation (p12). I suggest adding text along the lines of: '...,the majority of which resolved spontaneously within one year'.5) In the study limitations (p14, line 45) I suggest adding a
--

	sentence along the lines of: 'Furthermore, a degree of under-reporting is likely with any voluntary surveillance system'. 6) p14, line 38. Change wording from: '...the vast majority of cases in those <15 years who have been managed by these professionals.' to '...the vast majority of cases in those <15 years will have been managed by these professionals'. 7) The following text in the summary (p14, line 52) is somewhat misleading: 'One year after diagnosis, only 56% of children have made a full recovery without residual problems...'. The term residual suggests ongoing / unresolved complications, whilst the figure includes children with complications limited to the acute presentation which will have completely resolved. I suggest changing the wording to 'Only 56% of children made a full recovery with no acute or chronic complications or recurrent episodes of AP'
--	---

REVIEWER	Reviewer name: Dr. Paul D Losty Institution and Country: University of Liverpool, United Kingdom of Great Britain and Northern Ireland Competing interests: None
REVIEW RETURNED	09-May-2022

GENERAL COMMENTS	This study provides a qualitative narrative review of outcomes of UK racial group pediatric patients sustaining acute pancreatitis using a BPSU style survey. The findings are of thematic topical interest though do not provide any overarching or breaking news information for the BMJ. The study has significant and author acknowledged historical limitations being now conducted over some 8 years in 2013 or 2014 before Journal submission. Information on surgical management would be helpful in any future journal submissions.
--

VERSION 1 – AUTHOR RESPONSE

Response to Reviewers We thank reviewers and editors from BMJ Open Paediatrics for your comments. We have responded to the reviewer's comments below and have updated the manuscript to reflect the suggested changes. Editor in Chief Comments to Author: Introduction last para 1st sentence delete "the first" and replace with "a". Journal policy is to avoid describing studies as the first. Thank you for highlighting this. We have made the suggested change. Table 2 describes 90 cases, but the total is only 88. What happened to the missing two? Thank you for highlighting this. The two patients died in hospital – this has now been added to the table. Associate Editor Comments to the Author: Thank you for transferring this manuscript to BMJ Paediatrics Open and responding to the comments raised at previous peer review. It has been reviewed and please respond to the points raised. In addition the introduction should include background that these data relate to follow up from an earlier BPSU that was published - it is helpful for the reader if this is clear. The title could also reflect that this is follow up from a previous study. Thank you for your comments. We have made the suggested changes to the manuscript. Reviewer: 1 Dr. Paul D Losty, University of Liverpool Comments to the Author This study provides a qualitative narrative review of outcomes of UK racial group pediatric patients sustaining acute pancreatitis using a BPSU style survey. The findings are of thematic topical interest though do not provide any overarching or breaking news information for the BMJ. The study has significant and author acknowledged

historical limitations being now conducted over some 8 years in 2013 or 2014 before Journal submission. Information on surgical management would be helpful in any future journal submissions. Thank you for your comments. We have added detail on surgical management including a new table 2 describing the details of surgical management. Reviewer: 2 Dr. Emre Basatemur, Barts Health NHS Trust Comments to the Author This paper describes the clinical outcomes (initial and medium term to 1 year) from a national prospective surveillance study of acute pancreatitis in UK children conducted through the BPSU surveillance system (with additional reporting by paediatric surgeons affiliated with BAPS). The manuscript supplements previously published data from this study, which reported incidence rates, demographic factors, and reported aetiological associations. The paper reports original results, which are of clinical relevance and interest, and merits publication. The authors should be commended on achieving an impressive response rate for 1 year follow-up data (96%). Thank you for your comments. I have a few minor grammatical comments only: 1) There are a few references in the manuscript to 'long term' outcomes. One year followup would more appropriately be described as 'medium term' rather than long term. References to 'long term' should be changed to 'medium term'. E.g. in the short title (p1), 2nd line of the discussion (p12) Thank you for highlighting this. We have made the suggested change and removed references to "long term". 2) The text referring to the results in table 1 (page 8, outcomes subsection, 2nd sentence) could be written more clearly. I suggest something along the lines of: 'There was no association found between aetiology of AP and disease severity (p=0.22, Table 1).' Also, on p9 line 28, I suggest changing the text from 'The reported associations of AP...' to 'The reported aetiology of AP....' Thank you for your comment. We have made the suggested changes in the manuscript. 3) The authors report that 21% of cases required a surgical intervention. Do the authors have further information available regarding the types of surgical procedures performed? We have added detail on surgical management including a new table 2 describing the details of surgical management. 4) In the 1st paragraph of the discussion, the last sentence concerns pseudocyst formation (p12). I suggest adding text along the lines of: '...,the majority of which resolved spontaneously within one year'. Thank you for your comment. We have made this change to the manuscript. 5) In the study limitations (p14, line 45) I suggest adding a sentence along the lines of: 'Furthermore, a degree of under-reporting is likely with any voluntary surveillance system'. Thank you for your comment. We have made the suggested change in the manuscript. 6) p14, line 38. Change wording from: '...the vast majority of cases in those *who* have been managed by these professionals.' to '...the vast majority of cases in those *will* have been managed by these professionals'. Thank you for highlighting this. We have made the suggested change in the manuscript. 7) The following text in the summary (p14, line 52) is somewhat misleading: 'One year after diagnosis, only 56% of children have made a full recovery without residual problems...'. The term residual suggests ongoing / unresolved complications, whilst the figure includes children with complications limited to the acute presentation which will have completely resolved. I suggest changing the wording to 'Only 56% of children made a full recovery with no acute or chronic complications or recurrent episodes of AP' Thank you for highlighting this. We have made the suggested change in the manuscript.